# Sex-Specific Expression of Histone Lysine Demethylases (KDMs) in Thyroid Cancer

**DOI:** 10.3390/cancers16071260

**Published:** 2024-03-23

**Authors:** Leila Shobab, Hui Zheng, Kirk Jensen, Maria Cecilia Mendonca-Torres, Matthew McCoy, Victoria Hoperia, Jennifer Rosen, Leonard Wartofsky, Kenneth Burman, Vasyl Vasko

**Affiliations:** 1Department of Medicine, Division of Endocrinology, MedStar Washington Hospital Center, Washington, DC 20010, USA; 2Department of Surgery, MedStar Washington Hospital Center, Washington, DC 20010, USA; hui.zheng@medstar.net (H.Z.);; 3Department of Pediatrics, Uniformed Services University of the Health Sciences, Bethesda, MD 20814, USA; kirk.jensen@usuhs.edu (K.J.); vasyl.vasko@gmail.com (V.V.); 4Innovation Center for Biomedical Informatics, Georgetown University Medical Center, Washington, DC 20007, USA; 5Institute of Biology and Medicine, Kyiv National University, 02000 Kyiv, Ukraine; hoperiavictoria@gmail.com

**Keywords:** thyroid cancer, sex difference, KDMs, histone lysine demethylases

## Abstract

**Simple Summary:**

Thyroid cancer is the most common endocrine malignancy and is more frequently detected in women than in men. An understanding of the molecular features governing thyroid carcinogenesis in women and men may open avenues toward the development of sex-specific treatments for female and male patients with thyroid cancer. To this end, we performed an analysis of sex-biased genes in normal thyroids as well as benign and malignant thyroid tumors. We demonstrated that normal thyroid tissue has a sex-specific molecular signature, and the development of thyroid cancer is associated with the differential expression of sex-biased genes. In confirmatory studies, we demonstrated that the expression of histone lysine demethylases (KDMs) is strongly associated with sex. Together, our data demonstrate that the development of thyroid cancer may be associated with different sex-specific molecular changes. The sex-specific expression of KDMs, coupled with cancer-related alterations in their intracellular localization, may provide a clue for understanding the mechanisms underlying sex differences in thyroid tumorigenesis.

**Abstract:**

Background: The incidence of thyroid cancer in women is 3–4-fold higher than in men. To characterize sex-specific molecular alterations in thyroid cancer, we examined the expression of sex-biased genes in normal thyroids and thyroid tumors. Methods: Ingenuity pathways analysis was used to define sex-biased gene networks using data from the Cancer Genome Atlas (TCGA). Confirmatory studies were performed through the analysis of histone lysine demethylases (KDMs) expression by real-time PCR and immunostaining. Results: In normal thyroids, 44 sex-biased genes were comparatively upregulated in male and 28 in female patients. The expressions of 37/72 (51%) sex-biased genes were affected in cancer tissues compared with normal thyroids. Gene network analyses revealed sex-specific patterns in the expressions of KDM5C, KDM5D, and KDM6A. In confirmatory studies, KDM5D mRNA and protein were detected only in males, whereas KDM5C and KDM6A were detected in samples from male and female patients. Nuclear staining with anti-KDMs was found in normal thyroids, but a loss of nuclear expression with a concomitant gain of cytoplasmic staining was observed in cancer tissues. Conclusions: Normal thyroids have a sex-specific molecular signature, and the development of thyroid cancer is associated with a differential expression of sex-biased genes. The sex-specific expression of KDMs, coupled with cancer-related alterations in their intracellular localization, may contribute to mechanisms underlying sex differences in thyroid tumorigenesis.

## 1. Introduction

Thyroid cancer (TC) is the most common endocrine neoplasm [1,2], of which 94% are well-differentiated thyroid carcinomas (DTCs) of either the papillary (PTC) or follicular (FTC) type, with the PTC accounting for > 85% of diagnoses in the USA and the FTC reflecting 7% [2,3,4]. TC is a non-reproduction-based cancer with a strong female bias, with 3–4-fold higher incidence in women; however, males demonstrate more aggressive tumors with higher mortality [1,3,5,6,7].

The incidence rates of TC in women differ between regions of the world [8]. The highest TC incidence rate in the world was reported in several ethnic groups in New Caledonia, the South Pacific. The average annual rate of TC was 9.1/100,000 population. The highest rates were observed on the Island of Lifu (31.5/100,000). As commonly seen in other populations, the incidence was 7.8-fold higher in females than in males [9,10]. Together, epidemiological data show that intrinsic (sex hormones, aging, and genetic background) and extrinsic (environmental) factors may contribute to a sex-related dimorphism in TC [6,8,9,10].

The role of sex hormones in TC has been extensively examined previously. The expression of estrogen receptors (ERs) has been shown in TCs, and levels of ERs were found to be higher in cancers compared with normal thyroid tissue and benign nodules [11]. It has been shown that treatment by estrogen or agonists of ERs can enhance, whereas treatment with ER inhibitors decreases, the proliferation of the malignant thyroid cells [11,12]. It has also been demonstrated that specific patterns of ER and androgen receptor (AR) expressions were associated with a more aggressive phenotype of TC (10). However, no sex-specific differences have been established between the patterns of ER and AR expression in TC tissues from female and male patients [12].

While previous studies have suggested that sex differences in cancer may arise through the effect of circulating sex hormones, it has also been suggested that sex bias is derived from genetics and chromosomes independent of sex hormones. For example, the X-linked lysine demethylases 6A and 5C, as well as the Y-linked paralogs lysine demethylase 5D, may be regulators of the incidence and prognosis of sex-specific cancer [13,14]. Analysis of the molecular data on multiple cancer types using The Cancer Genome Atlas (TCGA) database provided an opportunity to address this problem [15]. TC was identified as a cancer with a strong sex-biased gene signature. Interestingly, in patients with TC, the patterns of somatic mutations were not different between females and males. In contrast, sex-specific signatures were established through the analysis of genes related to methylation and microRNA and mRNA expression. Analysis of normal thyroid tissue samples from female and male patients also revealed sex-specific patterns of mRNA expression.

While the role of histone lysine demethylases (KDMs) in the progression of prostate, ovarian, and breast cancers is partially understood, there is limited knowledge regarding potential sex-specific differences in the expressions of these enzymes in patients with thyroid cancer. Understanding such differences could shed light on the underlying molecular mechanisms contributing to thyroid cancer development and progression, potentially informing sex-specific diagnostic and therapeutic strategies.

We hypothesized that there are significant differences in the expression levels of histone lysine demethylases (KDMs) between male and female thyroid cancer patients, suggesting a potential sex-specific regulatory mechanism in thyroid cancer pathogenesis.

This study aimed to investigate whether sex-specific differences exist in the expressions of various KDMs within thyroid cancer tissues, providing insights into the potential role of epigenetic regulation in the sexual dimorphism observed in thyroid cancer. We employed an ingenuity pathways analysis (IPA) to define sex-biased gene networks in normal thyroids and thyroid tumors. Next, we focused further analysis on an IPA-defined network that included a set of sex-biased epigenetic regulators (histone lysine demethylases). Finally, we performed confirmatory studies through the analysis of mRNA and protein expressions in paired samples from normal thyroids and TCs from female and male patients.

## 2. Materials and Methods

### 2.1. Patients and Thyroid Tissue Samples

The protocol was approved by the local Institutional Review Board (IRB). This study was performed using data from the electronic medical records (EMRs) of patients in follow-up at MedStar Washington Hospital Center and MedStar Georgetown University Hospital, and thyroid tissue samples from tumors and corresponding normal thyroid tissues. Patients were screened with the inclusion criteria of a confirmed diagnosis of PTC or benign adenomas after total or near-total thyroidectomy. Patients’ data obtained from EMR included age, gender, initial presentation, histological subtype, TNM status, and the presence of loco-regional or distant metastasis. In total, tissue samples from 77 patients were examined (27 adenomas and 50 PTCs).

### 2.2. Computational Analysis

The UCSC Xena (https://xena.ucsc.edu/ (accessed on 1 January 2020) browser was used to explore functional genomic data sets for correlations between genomic and/or phenotypic variables. The genomic data and clinical–pathological characteristics of PTC patients were obtained from the TCGA.

Sex-biased gene networks and interactions were analyzed using ingenuity pathway analysis (IPA) software (https://digitalinsights.qiagen.com/IPA (accessed on 1 January 2020). Those genes with known gene symbols (HUGO) were uploaded into the software and mapped using the Ingenuity Pathways Knowledge Base. Networks of these genes were algorithmically generated based on their connectivity. The network identified was then presented as a graph indicating the molecular relationships between genes/gene products. Genes were integrated into the computationally generated networks on the basis of the evidence stored in the IPA knowledge memory indicating relevance to this network.

### 2.3. Quantitative Real-Time PCR (Q-RT-PCR)

Thyroid tissue samples were collected during surgery, immediately placed in RNA later solution, and stored at −80 °C for further use. Total RNA and DNA were extracted from RNA later-preserved tissue derived from paired tumors and normal samples from the same patient. Tissues were homogenized and followed by lysis with TissueLyser III (Qiagen, Inc., Valencia, CA, USA) according to the manufacturer’s protocol. Nucleic acids were purified with the AllPrep DNA/RNA Mini Kit (Qiagen, Inc., Valencia, CA, USA). DNA and RNA concentrations were measured with a Nanodrop 2000 spectrophotometer (Thermo Fisher Scientific, Inc., Waltham, MA, USA). Reverse transcription was performed using a SuperScript IV VILO Master Mix (Invitrogen Life Technology, Carlsbad, CA, USA) including no-RT control reactions for each experiment. The RT^2^ qPCR Primer Assay (Qiagen, Inc., Valencia, CA, USA) for KDM5C (NM_004187), KDM5D (NM_004653 and KDM6A (NM_021140), and 18S (NR_003286) was used for gene expression analyses by real-time PCR. Each sample was run in triplicate using a LightCycler^®^ 96 (Roche, New York, NY, USA) with a SYBR™ Green Universal Master Mix (ThermoFisher Scientific, Inc., Waltham, MA, USA). 18S (NR_003286) was used as a control for the amplification of target genes by real-time PCR. The relative fold gene expression of samples was calculated using the delta-delta Ct method.

### 2.4. Immunohistochemistry

Immunostaining was performed on commercially available human TC tissue microarray slides (US Biomax Inc., Rockville, MD, USA). KDM protein expression and localization by immunostaining were examined on 3 tissue microarray slides, each containing 80 thyroid tissue samples from 60 female and 20 male patients. On each slide, there were 10 samples from normal thyroid tissues, 20 from follicular cancer (FC), 44 from papillary thyroid cancer (PTC), and 6 from poorly differentiated thyroid cancer (PDTC). Sections were dewaxed, soaked in alcohol, and then microwave-treated in antigen unmasking solution (Vector Labs, Burlingame, CA, USA) for 10 min. Endogenous peroxidase activity was quenched by incubation in BLOXALL Blocking Solution for 10 min. The sections were then incubated for 20 min in 2.5% horse serum and incubated at 4 °C overnight with anti-KDM5C (SmcX Antibody (G-10): sc-376255), anti-KDM5D (SmcY Antibody (4C6): sc-293280), and anti-KDM6A (UTX Antibody (E-8): sc-514859). All primary antibodies were purchased from Santa Cruz Biotechnology, Santa Cruz, CA, USA.

Immunostaining was performed using the ImmPRESS® Horse Anti-Rabbit IgG PLUS Polymer Kit (Vector Labs, Burlingame, CA, USA) per the manufacturer’s instructions. Peroxidase staining was revealed with the ImmPact DAB peroxidase substrate kit (Vector Labs, Burlingame, CA, USA). Sections were counterstained with hematoxylin and mounted. The antiserum was omitted in the negative control.

Staining intensity (SI) was assessed according to a categorical scale: 0, no staining; 1, faint staining; 2, slight staining; 3, moderate staining; and 4, strong staining. The percentage of positively stained cells (PP) was assessed as 0, no positive cells; 1, 0–25% positive cells; 2, 26–50% positive cells; 3, 51–75% positive cells; and 4, 76–100% positive cells. An overall IHS score was calculated by multiplying the staining intensity (SI) by the percentage of positive staining or the staining frequency (PP) scores (range of possible scores: 0–16).

### 2.5. Statistical Analysis

Data were analyzed using SPSS 12 software. For non-categorical data, a two-tailed unpaired Student’s *t*-test was used. For categorical data, a two-tailed Fisher’s exact test was used. A *p*-value of < 0.05 was considered significant.

## 3. Results

### 3.1. Ingenuity Pathway Analysis of Sex-Biased Genes in Normal Thyroids

We interrogated the TCGA Thyroid Cancer (THCA) database and examined the expression of 72 genes with known sex-biased expression (11) using the UCSC Xena and Ingenuity Pathway Analysis (IPA) platforms. The information on 72 sex-biased genes in normal thyroids is summarized in Appendix A. An interrogation of TCGA data revealed that 44 genes were upregulated in normal male thyroid tissue compared with normal female thyroid tissue (Figure 1A). In contrast, the expressions of 28 genes were increased in female normal compared with male normal thyroid tissue.

The list of 72 differentially expressed genes in normal thyroids from female and male patients was uploaded into the Ingenuity Pathway Analysis software. A “core analysis” in the IPA software was performed to characterize sex-biased gene networks and their associated functions, as well as common biological processes, diseases, and canonical signaling pathways. The biological process related to the expression of sex-biased genes included embryonic development and the development of the endocrine system. The expression of sex-biased genes was associated with developmental and genetic disorders (Klinefelter syndrome, Turner syndrome, and JARID1C-related X-linked intellectual disability), endocrine system disorders, and reproductive diseases. Genes that were upregulated in males compared with females play a role in spermatogenesis, the differentiation of mesodermal cells, and the development of the genitourinary system, as well as prostate cancer. In contrast, genes that are upregulated in females compared with males have been demonstrated in the development of ovarian and breast cancers. An overlay of sex-biased gene networks with canonical signaling pathways demonstrated that these genes are involved in the regulation of eIF4 and p70S6K signaling, the regulation of eukaryotic translation initiation, and the response of EIF2AK4 (GCN2) to amino acid deficiency.

IPA also identified five significant networks associated with the differentially expressed genes in the female and male normal thyroid. These networks were scored based on the number of genes participating in any particular network. The most significant network included 22 sex-biased genes (Figure 1B). The IPA suggested the role of sex-biased genes in histone modification. Specifically, a node converging on histone-3 methylation included a set of demethylases (KDM5C, KDM5D, and KDM6A) that contribute to the formation of histone-modifying complexes and catalyze the demethylation of tri/di-methylated histone H3.

### 3.2. Ingenuity Pathway Analysis of Sex-Biased Genes in Thyroid Cancer

After the characterization of sex-biased genes in normal female and male thyroids, we, next, compared the expressions of these genes in normal and cancer tissues. Comparative analysis of the sex-biased genes demonstrated that the expressions of 37/72 (51%) sex-biased genes were affected in cancers compared with normal thyroid tissue (Figure 2A). The expressions of 12 sex-biased genes (COPG2, HDHD1A, INPP1, ITGA5, LCN6, MYO1A, RAET1L, RPS4X, TAGLN2, VAX2, ZRSR2, and ZYX) were increased in cancer tissue compared with normal thyroid tissue (Appendix A), and the expressions of 25 genes (ABHD10, ACRBP, ACTR3C, AGPAT3, AMELY, APITD1, CWH43, DDX3X, EIF1AX, EPHB2, IL12A, KDM6A, PCDH11Y, PIM1, PNPLA4, PPP1R2P9, PRKY, PTCD2, RBM46, SEPHS1, SYAP1, TRAM1L1, TXNDC15, ZNF396, and ZNF709) was decreased (Appendix A).

In male patients, the comparison of sex-biased genes in normal and cancer tissues showed a shift toward a “female-like” gene signature in cancers compared with normal tissues. The expressions of 18 out of 44 genes with intrinsically high levels in male normal thyroids were decreased in male cancer tissue. In contrast, the expressions of 10 out of 28 genes with low baseline levels in male normal thyroid tissue were increased in the TCs of males. In female patients, the comparison of sex-biased genes in normal and cancer tissues showed that the “female-like” gene signature became even more accentuated in cancer tissue. The expressions of 9 out of 28 genes with intrinsically high levels in normal female thyroids were increased in female TCs. The expressions of 15 out of 44 genes with low levels in female normal thyroid tissue compared with male normal thyroids were even further decreased in cancers of females.

We also examined the associations of TC sex-biased genes with cellular functions and diseases. An IPA analysis of molecular and cellular functions demonstrated the role of sex-biased genes in cell-to-cell signaling, the cell cycle, and cellular assembly and organization. We next investigated the interactions between sex-biased genes and the associations of these genes with canonical signaling pathways. We identified four networks associated with the differentially expressed sex-biased genes in normal thyroid and TC tissues. The most significant network included 19 sex-biased genes (Figure 2B). An overlay of sex-biased gene networks with canonical signaling pathways demonstrated that these genes are involved in the regulation of ERK/MAPK, AKT, and integrin signaling, as well as interleukin signaling. A comparison of sex-biased genes in normal and TC tissues also suggested the role of epigenetic mechanisms and histone-3 methylation in thyroid carcinogenesis.

Together these data show that normal thyroid tissue has a sex-specific molecular signature, and the development of TC in female and male patients is associated with the differential expression of sex-biased genes. Specifically, the development of TC in males was associated with a shift in the expression of sex-biased genes toward a “female-like” genotype. Analysis of gene networks demonstrated the role of sex-biased genes in the regulation of well-established TC drivers such as ERK/MAPK and AKT signaling and indicated the role of histone lysine demethylases (KDMs) and histone methylation epigenetic regulators in the development of TC.

### 3.3. Computational Analysis of Histone Lysine Demethylases (KDMs) in Thyroid Cancer

A comprehensive analysis of molecular differences between male and female patients demonstrated a strong effect of gender on gene methylation in TCs (11). Therefore, we further examined sex-biased histone lysine demethylases (KDM5C, KDM5D, and KDM6A), which have an established role in epigenetic regulation. As demonstrated in Figure 3, KDMs were differentially expressed in TC tissues from female and male patients. While female TC tissue demonstrated increased mRNA levels of KDM5C and KDM6A, KDM5D was exclusively expressed in males and was overexpressed in male TCs compared with normal thyroid tissue (Figure 3).

We then employed an IPA analysis to interrogate the direct upstream regulators as well as downstream targets for these KDMs (Figure 4).

This analysis suggested the role of KDMs in the regulation of ERK/MAPK signaling as well as histone-3 methylation (Figure 4). Interrogation of the IPA data revealed that KDMs contribute to the formation of WDR5-containing histone-modifying complexes and play a role in histone demethylase activity, metal ion binding, and oxidoreductase activity. KDMs are also involved in the regulation of chromatin organization, regulation of transcription, androgen receptor signaling pathway, as well as T-cell antigen processing and presentation.

We also utilized the IPA knowledgebase to derive insight from previously reported findings related to the involvement of KDMs in TC. Interrogation of the IPA revealed that mutations in KDM5C gene (c.151-339C>T), KDM5D gene (c.1426G>T translating to p.E476* [somatic nonsense]), and KDM6A gene (substitution c.*168A>C [somatic]) have been observed in human TCs.

### 3.4. Analysis of KDM mRNA Expression in Normal Thyroids and Benign and Malignant Thyroid Tumors

To clarify the role of KDMs in thyroid tumorigenesis, we next examined the mRNA expressions of KDM5C, KDM5D, and KDM6A by RT-PCR in a set of normal thyroid and tumor tissue samples from 77 patients, who underwent surgery for thyroid nodules. Histopathological analysis revealed benign tumors in 27 cases (16 females and 11 males) and TC in 50 cases (37 females and 13 males).

We first compared KDM expression in normal female and normal male thyroid tissues. The mRNA level of KDM5C was 2.67-fold higher in females compared with male normal thyroid tissue (*p* = 0.0029). Consistent with TCGA data, KDM5D was detected only in male patients. There were no significant differences between the mRNA levels of KDM6A in normal thyroid tissues from female and male patients (fold changes: 1.1; *p* = 0.3457).

Next, we analyzed KDM expression in tissue samples from 16 patients who underwent surgery for benign thyroid nodules and 50 patients operated on for TC. The mRNA levels of KDMs in tumors were compared with those observed in the corresponding normal thyroid tissues. The threshold value of at least three-fold differences between tumors and the corresponding normal tissues was selected to classify cases as tumors with increased, decreased, or unchanged levels of KDMs. The results of the RT-PCR analysis of KDM5C, KDM5D, and KDM6A in tumors diagnosed in female and male patients are summarized in Table 1 and Table 2.

Analysis of KDM mRNA and clinicopathological characteristics indicated that the levels of KDM5C, KDM5D, and KDM6A expression were not significantly associated with the patient’s age, tumor size, the presence of lymph node metastases. or extra-thyroidal extension. For 23 patients with TCs, molecular data were available. Mutations in BRAF and RAS oncogenes were detected in 15 and 5 cases, respectively, and in 3 cases, TCs harbored gene fusions (1 PPARg/PAX8 and 2 NTRK3 fusions). There was no significant association between the levels of KDM expression and thyroid oncogene mutation status.

### 3.5. Immunohistochemical Analysis of KDM Expression in Thyroid Cancers

We examined KDM protein expression and localization by immunostaining on tissue microarray slides containing 80 thyroid tissue samples from 60 female and 20 male patients. On each slide, there were 10 samples from normal thyroid tissues, 20 from follicular cancers (FCs), 44 from papillary thyroid cancers (PTCs), and 6 from poorly differentiated thyroid cancers (PDTCs). The results of immunostaining with anti-KDMs in tissue samples from female patients are summarized in Table 3.

KDM5C was detected in normal thyroid tissues as well as cancer tissue samples, and no significant differences were found between immunostaining scores in FTC, PTC, or PDTC tissues and normal thyroid tissues. However, the intracellular localization of KDM5C was different in normal thyroid tissues and thyroid tumors. While exclusively nuclear expression of KDM5C was detected in all cases of normal thyroids, loss of nuclear KDM5C was frequently observed in TCs (Figure 5).

In FTC, PTC, and PDTC tissues, only nuclear expression of KDM5C was found in 6/14 (42.8%) of FTC, 16/38 (42.1%) of PC, and 2/4 (50%) of PDTC tissues. Loss of nuclear KDM5C was associated with increased KDM5C expression in the cytoplasm of tumor cells. Cytoplasmic localization of KDM5C was observed in 5/14 (35.7%) of FTC, 12/38 (31.5%) of PTC, and 0/2 (0%) of PDTC tissues. In 15 cases (3 FTCs, 10 PTCs, and 2 PDTCs), there was no detectible KDM5C staining in the examined cancer samples.

KDM5D was not detected by immunostaining in normal thyroid or cancer tissue samples from female patients.

KDM6A was detected in normal thyroid as well as cancer tissue samples. There were no significant differences between immunostaining scores in normal thyroid tissues and PTCs. However, immunostaining scores in normal thyroid tissues were significantly higher than in FTCs (*p* = 0.0015) and PDTCs (*p* = 0.0048). Analysis of KDM6A intracellular localization showed exclusively nuclear expression in 4/4 (100%) of normal thyroid tissues, 3/14 (21.4%) of FTCs, 8/38 (21.5%) of PTCs, and 0/4 (0%) of PDTCs. In contrast, cytoplasmic or mixed nuclear cytoplasmic expression was detected in 0/4 (0%) of normal thyroid tissues, 11/14 (78.6%) of FTCs, 29/38 (76%) of PTCs, and 4/4 (100%) of PDTCs. In one case of PTC, KDM6A expression was not detected.

Next, we analyzed KDMs in normal thyroid tissues and TCs from male patients, and the results of immunostaining with anti-KDMs in the tissue samples from male patients are summarized in Table 4.

KDM5C was detected in normal thyroid as well as cancer tissue samples, and no significant differences were found between immunostaining scores in normal thyroid tissues, FTCs, PTCs, or PDTC. The intracellular localization of KDM5C was exclusively nuclear in normal thyroid tissues. In TCs, nuclear KDM5C was detected in 3/6 (50%) of FTCs, 1/6 (16.6%) of PTCs, and 1/2 (50%) of PDTCs.

The immunostaining scores for KDM5D were significantly higher in normal thyroid tissues compared with FTCs (*p* = 0.0247), PTCs (*p* = 0.0004), and PDTCs (*p* = 0.0.0004). Nuclear KDM5D expression was found in all normal thyroid tissue samples. In cancers, nuclear KDM5D was detected in 1/6 (16.6%) of FTCs, 2/6 (33.3%) of PTCs, and 0/2 (0%) of PDTCs. Loss of nuclear KDM5D was associated with increased KDM5D expression in the cytoplasm of tumor cells (Figure 6).

KDM6A expression was detected in all examined samples with no significant differences between immunostaining scores in normal thyroid tissues, FTCs, PTCs, and PDTCs. Nuclear KDM6A expression was found in all normal thyroid tissue samples. In cancers, nuclear KDM6A was detected in 2/6 (33.3%) of FTCs, 3/6 (50%) of PTCs, and 0/2 (0%) of PDTCs. Loss of nuclear KDM6A expression was associated with increased KDM6A expression in the cytoplasm of tumor cells.

## 4. Discussion

Thyroid cancer exhibits a striking sex bias in its incidence, with a notably higher prevalence among females. Recent advancements in genomic studies have shed light on the role of sex-biased genes in thyroid tumorigenesis, indicating potential underlying mechanisms contributing to this gender disparity. In this context, an understanding of the specific genes and molecular pathways that display sex-specific effects in TC could unravel crucial insights into disease development in female and male patients and may contribute significant insight to the development of sex-specific, targeted, and more effective strategies for the prevention, diagnosis, and treatment of TC.

Our study provides an exploration of sex-biased gene networks in TC, with an analysis of histone lysine demethylases (KDMs) in normal thyroid tissues, benign thyroid tumors, and TCs.

First, our interrogation of the TCGA database demonstrated a robust sex-biased signature in normal thyroid tissues with differential expression of sex-biased genes between male and female patients, in keeping with previous studies [15]. Subsequent analysis of the gene networks and signaling pathways demonstrated that sex-biased genes are involved in cellular proliferation through the regulation of ERK/MAPK and p70S6K signaling and have a role in epigenetic regulation. Interestingly, a normal thyroid in females was characterized by enrichment in genes belonging to the interleukin family, specifically IL-12. IL-12 plays an important role in the activities of natural killer cells and T lymphocytes and mediates the enhancement of the cytotoxic activity of NK cells and CD8+ cytotoxic T lymphocytes. IL-12 demonstrates potent antitumor activity by enhancing the Th1/Th2 response, facilitating cytotoxic T cell (CTL) recruitment into tumors, inhibiting tumor angiogenesis, and depleting immunosuppressive cells in the tumor microenvironment (TME). Our findings support recent evidence indicating a critical role of KDM6A (which encodes the UTX protein) in controlling NK cell development and a function in regulating chromatin accessibility and gene expression vital for NK cell homeostasis and effector function [16]. These data suggest that even at baseline conditions in the normal thyroid gland, the immunological microenvironment is distinct between males and females, as previously established in studies on both innate and adaptive immune responses [17,18]. Further analysis of sex-biased genes that were upregulated in cancer tissues compared with normal thyroid tissues revealed that these genes were associated with the ERK/MAPK, AKT, integrin, actin cytoskeleton, and p70S6K signaling pathways. In female patients, the development of cancer was associated with increased expression of genes regulating the immunological microenvironment. Specifically, retinoic acid early transcript 1L (RAET1L) was upregulated only in female cancer compared with normal tissue. The RAET1L (ULBP6) gene has been shown to encode the ligand for the immunoreceptor NKG2D, an important mediator of anti-tumor activity [19]. Studies on RAETL1 in thyroid cells demonstrated that anaplastic cancer cell lines are sensitive to NK cell-mediated lysis via ULBP2/5/6 and chemoattractant CXCR3-positive NK cells, suggesting that patients with ATC may benefit from NK cell-based immunotherapy [20]. Our results demonstrate the differential expression of sex-biased genes contributing to the immuno-microenvironment in normal thyroid tissue as well as in TC and suggest that NK cell-based immunotherapy may have sex-specific efficacy in patients with advanced TC refractory to standard therapy.

In TC, DNA methylation and histone modification constitute an important mechanism that underlies the epigenetic regulation of gene transcription [21,22]. However, sex-specific patterns of epigenetic regulators in TC have not been reported. In our study, IPA analysis of sex-biased gene networks suggested the role of histone lysine demethylases (KDM5C, KDM5D, and KDM6A) in TC. By performing real-time PCR of normal thyroid tissues, we found that KDM5C showed significantly higher expression in females, KDM5D was exclusively detected in male patients, and no significant sex differences were observed for KDM6A.

Histone lysine demethylases (KDMs) are enzymes involved in removing the methyl groups on lysine residues in nucleosomes. This can lead to either the activation or suppression of transcription, depending on which lysine residue is targeted. Therefore, the role of KDMs can either be oncogenic or tumor suppressive, contingent on the cell context and the enzyme isoform expressed [23,24,25,26]. The lysine methylation sites are found on histone H3 lysine K4, K9, K27, K56, and K79 and histone H4 lysine K20, allowing the mechanism for the regulation of a large array of biological processes [27,28].

The inactivation of tumor suppressor genes, such as KDMs, has been implicated in the pathogenesis of several cancers [26,29]. One mechanism implicated in the deactivation of several tumor suppressors is their nuclear export into the cytoplasm. Dysregulation of nuclear export can result in pathological conditions, including cancer [30,31]. Immunohistochemical analysis of KDMs showed sex-specific as well as tumor-specific patterns of KDM protein expression. The patterns of KDM protein intracellular localization were different in normal thyroid tissues (intra-nuclear localization) and TCs (common loss of nuclear staining with concomitant gain of cytoplasmic staining).

The intracellular localization of transcription factors or activated signaling molecules is an important factor underlying the biological behavior of cancer cells. In our previously published studies, we explored the effects of intracellular trafficking of activated AKT and demonstrated that the nuclear translocation of phosphorylated AKT was associated with the nuclear exclusion of p27, a well-characterized tumor suppressor. In addition, nuclear phospho-AKT expression was associated with increased migratory abilities of thyroid cancer cells [32,33]. In this context, not only the levels of KDM mRNA expression but also the intracellular localization of KDM proteins should be taken into consideration while analyzing the role of KDMs in the regulation of sex-specific features in TCs.

Chromosome region maintenance 1 (CRM1) is a nuclear receptor that recognizes proteins bearing a leucine-rich nuclear export sequence (NES) and is the most widely studied nuclear exportin that mediates the nuclear export of many tumor suppressor proteins. CRM1 overexpression has been implicated as an underlying mechanism in the pathogenesis of several cancers [30,34,35,36]. Inhibition of CRM1 has shown benefits in slowing down cancer growth and overcoming drug resistance to chemotherapeutic agents in several cancers including anaplastic thyroid cancer (ATC) [37,38,39,40,41,42]. The role of these mechanisms in the regulation of KDM intracellular trafficking as well as a detailed characterization of the downstream targets of KDMs and the biological role of KDM effectors need to be examined.

## 5. Conclusions

Normal thyroid tissue has a sex-specific molecular signature, and the development of thyroid cancer is associated with the differential expression of sex-biased genes. TC initiation and progression are regulated through multiple genetic and epigenetic mechanisms, including the methylation of histones. Further clarification of sex-specific epigenetic changes in TC will be important in understanding the sex-specific clinical features of TC, the identification of sex-specific molecular biomarkers, and the design of effective sex-specific targeted therapies. Our results provide new insights into the complex interplay of sex-biased genes and epigenetic regulators in TC, paving the way for a deeper understanding of the molecular mechanisms underlying sex-specific differences in thyroid tumorigenesis.

## Figures and Tables

**Figure 1 cancers-16-01260-f001:**
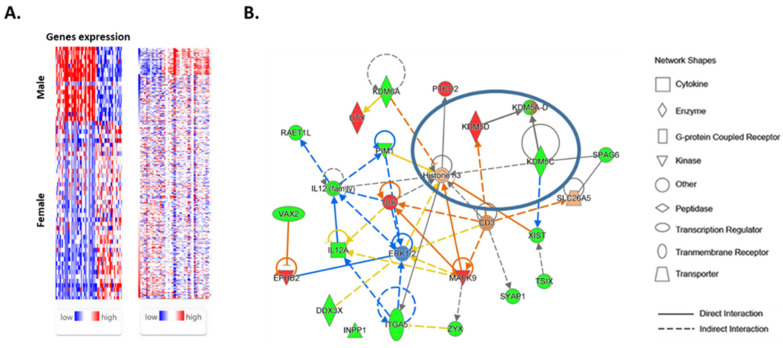
Expressions of sex-biased genes in normal female and male thyroid tissues: (**A**) Heatmap of 72 sex-biased genes showing their expression levels in normal thyroid tissue; 44 genes were upregulated in males and 28 genes in females. (**B**) IPA indicated a regulatory node converging on histone 3 methylation that included a set of demethylases (KDM5C, KDM5D, and KDM6A) that contribute to the formation of histone-modifying complexes suggestive of an important regulator in thyroid physiology. Red and green colors indicate genes that were up-regulated in male and female thyroid tissues, respectively. Blue color indicates a predictive inhibitory effects.

**Figure 2 cancers-16-01260-f002:**
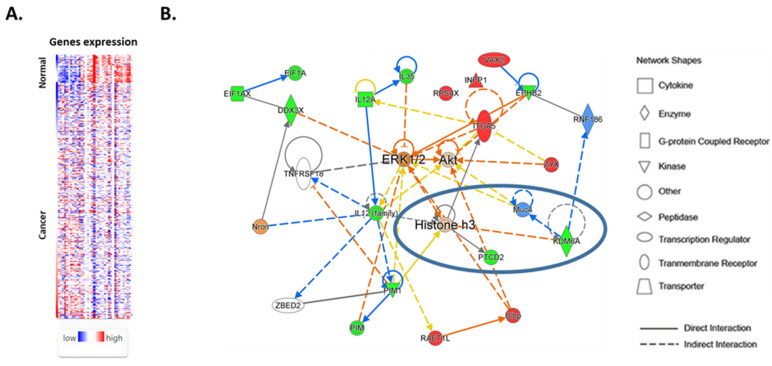
(**A**) Expressions of sex-biased genes in normal thyroid tissue and thyroid cancer. Expression levels of 37/72 (51%) of the sex-biased genes were altered in thyroid cancer tissue compared with paired normal tissue. (**B**) IPA indicated a regulatory node converging on ERK1/2, AKT, as well as on histone H3. Red and green colors indicate genes that were up-regulated in male and female thyroid tissues, respectively. Blue color indicates a predictive inhibitory effects.

**Figure 3 cancers-16-01260-f003:**
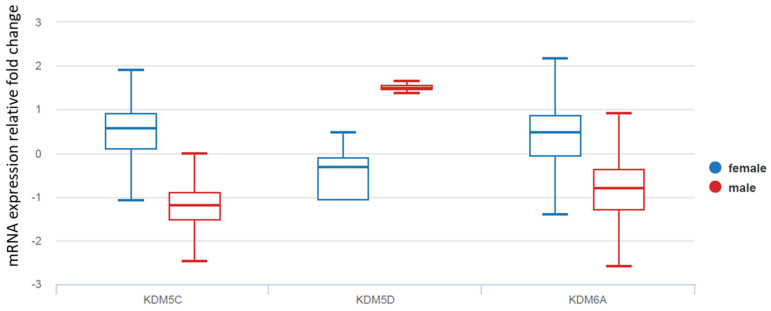
The comparative analysis of mRNA levels of KDM5C, KDM5D, and KDM6A in thyroid cancers from female and male patients based on TCGA database. Welch’s *t*-test confirmed over-expression of KDM5C in female TC [*p* = 0.000 (t = 26.3)]; overexpression of KDM5D in male TC [*p* = 2.440 × 10^−208^ (t = −58.48)]; and overexpression of KDM6A in female TC [*p* = 0.000 (t = 16.55)].

**Figure 4 cancers-16-01260-f004:**
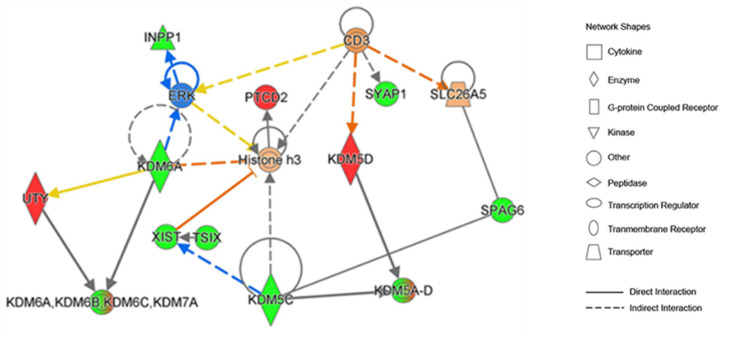
KDM gene network with upstream regulators and downstream effectors: IPA indicating important role of KDMs in regulation of ERK/MAP signaling as well as histone-3 methylation. Interrogation of IPA knowledgebase revealed links between KDMs and thyroid cancer. Red and green colors indicate genes that were up-regulated in male and female thyroid tissues, respectively. Blue color indicates a predictive inhibitory effects.

**Figure 5 cancers-16-01260-f005:**
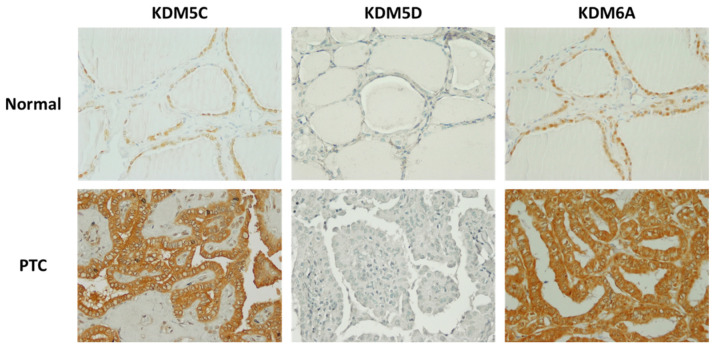
Immunohistochemical staining of KDM5C, KDM5D, and KDM6A in normal thyroid tissues and thyroid cancers from female patients. KDM5D was not detected in normal and cancer tissues of female patients. In normal thyroid tissues of females, both KDM5C and KDM6A were detected inside the nucleus, whereas in cancer tissues, both were detected primarily inside the cytoplasm. Magnification: ×400.

**Figure 6 cancers-16-01260-f006:**
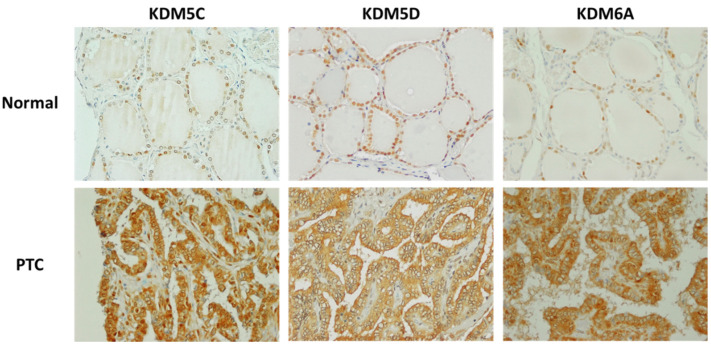
Immunohistochemical staining of KDM5C, KDM5D, and KDM6A in normal thyroid tissues and thyroid cancers from male patients: KDM5C, KDM5D, and KDM6A were all detected inside the nuclei of normal thyroid tissues, whereas in thyroid cancers, all three were primarily detected inside the cytoplasm. Magnification: ×400.

**Table 1 cancers-16-01260-t001:** Relative mRNA expression of KDMs in tumors versus corresponding normal thyroid tissues in female patients.

Genes	Benign			Malignant		
	Up	Down	No Changes	Up	Down	No Changes
KDM5C	0/16 (0%)	4/16 (25%)	12/16 (75%)	7/37 (20%)	2/37 (5%)	28/37 (75%)
KDM5D	ND	ND	ND	ND	ND	ND
KDM6A	1/16 (6%)	2/16 (13%)	13/16 (81%)	6/37 (16%)	2/37 (5%)	29/37 (79%)

ND: non detected.

**Table 2 cancers-16-01260-t002:** Relative mRNA expression of KDMs in tumors versus corresponding normal thyroid tissues in male patients.

Genes	Benign			Malignant		
	Up	Down	No Changes	Up	Down	No Changes
KDM5C	1/11 (9%)	0/11 (0%)	10/11 (91%)	1/13 (7%)	0/13 (0%)	12/13 (93%)
KDM5D	2/11 (18%)	1/11 (9%)	8/11 (73%)	2/13 (15%)	0/13 (0%)	11/13 (85%)
KDM6A	2/11 (18%)	2/11 (18%)	7/11 (64%)	0/13 (0%)	0/13 (0%)	13/13 (100%)

**Table 3 cancers-16-01260-t003:** Expression of KDM proteins in thyroid tissue samples from female patients.

Histology	Immunostaining Score		
	KDM5C (Average ± SD)	KDM5D (Average ± SD)	KDM6A (Average ± SD)
Normal	2.25 ± 1.2	0	9.75 ± 1.5
FTC	1.92 ± 2.0	0	5.1 ± 2.9
PTC	3.0 ± 3.2	0	10.7 ± 3
PDTC	2.0 ± 2.7	0	6.25 ± 2

**Table 4 cancers-16-01260-t004:** Expression of KDM proteins in thyroid tissue samples from male patients.

Histology	Immunostaining Score		
	KDM5C (Average ± SD)	KDM5D (Average ± SD)	KDM6A (Average ± SD)
Normal	1.5 ± 1.3	10.0 ± 2.4	7.0 ± 1.5
FTC	1.5 ± 1.5	4.7 ± 4.1	6.2 ± 4.5
PTC	1.7 ± 1.4	2.3 ± 2.7	7.5 ± 2.8
PDTC	1.5 ± 0.7	2.0 ± 0	7.0 ± 1.4

## Data Availability

The data presented in this study are available upon request from the corresponding author.

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
