# Peer review of "Sex-Specific Expression of Histone Lysine Demethylases (KDMs) in Thyroid Cancer"

_cancers, 2024, doi:10.3390/cancers16071260_

Round 1

Reviewer 1 Report

Comments and Suggestions for Authors

This manuscript investigates the sex-specific aspects of thyroid cancer (TC) by exploring gene expression. The study utilizes TCGA data to identify sex-biased gene networks in normal thyroid and TC, revealing differential expression of genes. In TC, upregulated genes are associated with signaling pathways and immunological regulation, suggesting potential benefits of NK cell-based immunotherapy, particularly in female patients. The study also focuses on histone lysine demethylases (KDMs) (KDM5C, KDM5D, and KDM6A, indicating sex-specific expression patterns and distinct intracellular localization in TC. In conclusion, the research underscores the importance of understanding sex-specific gene expression and epigenetic regulation in TC, providing insights for developing sex-specific targeted therapies and biomarkers. To strengthen the manuscript, these following aspects could be considered:

Any correlation with clinical data to strengthen the relevance of the findings to thyroid cancer patients.

The manuscript identified three KDMs based on the analysis. However, how does these enzymes contribute to thyroid cancer development and progression?

The authors should explore the correlations between the identified sex-biased genes and patient outcomes. For example, how the identified sex-specific differences may relate to clinical outcomes, prognosis, survival, or response to therapies?

Include functional assays to explore the effects of KDMs (inhibitors or genetic manipulation) on TC cells.

Considering the possible epigenetic changes, any available ChIP-seq data for validation?

Have the authors compared the expression of KDMs in TC from male and female patients? Any distinct pattern?

Minor:

Please conduct a thorough proofreading and formatting review. For example, extra spaces were spotted. Some spelling errors and extra words, including Line 158, regulatoryy. Line 442, conclusion.

It would be beneficial to explain the arrow, the line, the shapes, and the colors in networks from IPA analysis in Figure 1b, 2b , and 4a.

Author Response

This manuscript investigates the sex-specific aspects of thyroid cancer (TC) by exploring gene expression. The study utilizes TCGA data to identify sex-biased gene networks in normal thyroid and TC, revealing differential expression of genes. In TC, upregulated genes are associated with signaling pathways and immunological regulation, suggesting potential benefits of NK cell-based immunotherapy, particularly in female patients. The study also focuses on histone lysine demethylases (KDMs) (KDM5C, KDM5D, and KDM6A, indicating sex-specific expression patterns and distinct intracellular localization in TC. In conclusion, the research underscores the importance of understanding sex-specific gene expression and epigenetic regulation in TC, providing insights for developing sex-specific targeted therapies and biomarkers. To strengthen the manuscript, these following aspects could be considered:

  • Any correlation with clinical data to strengthen the relevance of the findings to thyroid cancer patients.

We analyzed expression of sex-biased genes and pathological findings such as tumor size and presence of lymph node metastases at the time of surgery using TCGA data base.

In male patients, the number of sex-biased genes that were downregulated in cancer correlated with tumor size. In T1, T2, T3 and T4 male thyroid cancers, the numbers of downregulated genes were 15, 17, 22 and 24, respectively. In addition, in male patients an expression of 15 sex-biased genes were decreased in N1 versus N0.

In female patients, analysis of sex-biased genes in function of tumor size demonstrated that in T1, T2, T3 and T4 tumors the number of downregulated genes were 21, 22, 26 and 25, respectfully.  Analysis of gender-biased genes in female thyroid cancer presenting with and without lymph node metastases showed that 12 genes were downregulated in N1 comparing to N0 cancers.

Together, these data suggested the potential tumor and/or metastases suppressive role of sex-biased genes role in thyroid cancer. 

  • The manuscript identified three KDMs based on the analysis. However, how does these enzymes contribute to thyroid cancer development and progression?

Aberrant histone lysine methylation that is controlled by histone lysine (K)-specific demethylase family of genes (KDMs) has been demonstrated in breast, prostate, ovarian cancers. The role of KDMs in development of thyroid cancer is unknown.

To provide experimental evidences confirming differential expression of sex-biased genes in female and male thyroid cells we examined KDMs expression in thyroid cancer cells lines that were established from female and male patients. We found that expression of KDM5D was detected only in KTC1 and C643 cells (derived from male-patient), but not in BCPAP and SW1736 cells, that derived from female patient. We believe that these cell lines could be used as a model for the exploration of the sex-specific function of KDMs in thyroid cancer in our further experiments.

  • The authors should explore the correlations between the identified sex-biased genes and patient outcomes. For example, how the identified sex-specific differences may relate to clinical outcomes, prognosis, survival, or response to therapies?

To examine sex-specific patterns of gene expression we conducted analysis on preselected homogenous groups of male and female patients with papillary thyroid cancers.  Patients presenting with poorly differentiated, anaplastic carcinomas were excluded, and as result, an excellent response to treatment was documented in most of patients. There is a need in further study specifically exploring the role of sex-biased genes in clinically aggressive thyroid cancer.

  • Include functional assays to explore the effects of KDMs (inhibitors or genetic manipulation) on TC cells.

We demonstrated expression of KDM5D in KTC1 and C643 cells (derived from male-patient), but not in BCPAP and SW1736 cells, that derived from female patient. We believe that these cell lines could be used as a model for the exploration of the sex-specific function of KDMs in thyroid cancer and can be used for gene-silencing/overexpression experiments as well as for the treatment with specific KDM inhibitors.  

  • Considering the possible epigenetic changes, any available ChIP-seq data for validation?

For identification of KDMs downstream targets the ChIP-seq experiments will be performed in our further studies.

  • Have the authors compared the expression of KDMs in TC from male and female patients? Any distinct pattern?

On mRNA and protein levels the expression of KDM5D was detected only in male thyroid tissue samples, but not in female tissue samples. KDM5C protein expression was higher in thyroid samples from female as compared to male patients [average IHS score 2.6 vs 1.5 (p=0.0251)]. KDM6A protein expression was higher in female as compared to male patients [average IHS score. 9 vs 7.1 (p=0.0239)].

Minor:

Please conduct a thorough proofreading and formatting review. For example, extra spaces were spotted. Some spelling errors and extra words, including Line 158, regulatoryy. Line 442, conclusion.

Thank you for bringing these errors to our attention. These errors have been now corrected and the manuscript has been thoroughly assessed for any additional spelling mistakes and typos and these have been corrected.

It would be beneficial to explain the arrow, the line, the shapes, and the colors in networks from IPA analysis in Figure 1b, 2b , and 4a.

Corrections were made in Figures 1, 2 and 4

Reviewer 2 Report

Comments and Suggestions for Authors

Overview

The current manuscript entitled “Sex-Specific Expression of Histone Lysine Demethylases 2 (KDMs) in Thyroid Cancer” represents an investigation into the role of sex-specific molecular alterations in thyroid cancer, focusing particularly on the expression of sex-biased genes and histone lysine demethylases (KDMs). The rationale and objectives are fairly clear. The statistical analysis and reporting are adequate. The figures and tables are informative. However, there are issues with the overall presentation.

General comments and questions

In the introduction, a more explicit statement is needed to highlight the knowledge gap and the hypotheses being tested in this study.

A statement of the rationale of why these particular KDMs (KDM5C, KDM5D, and KDM6A) were selected for detailed analysis over others is needed.

While the authors identified sex-specific expressions and localizations of KDMs, they may go and discuss the potential biological mechanisms being affected.

A more in-depth discussion needed, where more comparisons need to be made with existing literature. For example, highlighting how this study's findings align with or differ from previous research and what these differences might suggest about the underlying biology. Also, the discussion sometimes jumps between topics without clear transitions, making it challenging to follow the argument's thread.

Reference numbers should be placed in square brackets throughout the manuscript.

Specific comments and questions

Please remove the extra period in the end of the title.

Unexpected space in line 53, 58, 425 and more…

Author Response

The current manuscript entitled “Sex-Specific Expression of Histone Lysine Demethylases 2 (KDMs) in Thyroid Cancer” represents an investigation into the role of sex-specific molecular alterations in thyroid cancer, focusing particularly on the expression of sex-biased genes and histone lysine demethylases (KDMs). The rationale and objectives are fairly clear. The statistical analysis and reporting are adequate. The figures and tables are informative. However, there are issues with the overall presentation.

General comments and questions

  • In the introduction, a more explicit statement is needed to highlight the knowledge gap and the hypotheses being tested in this study.

Knowledge Gap:

While the role of histone lysine demethylases (KDMs) in progression of prostate, ovarian and breast cancers is partially understood, there is limited knowledge regarding potential sex-specific differences in the expression of these enzymes in patients with thyroid cancer. Understanding such differences could shed light on the underlying molecular mechanisms contributing to thyroid cancer development and progression, potentially informing sex-specific diagnostic and therapeutic strategies.

Hypotheses:

There are significant differences in the expression levels of histone lysine demethylases (KDMs) between male and female thyroid cancer patients, suggesting a potential sex-specific regulatory mechanism in thyroid cancer pathogenesis.

The study aimed to investigate whether sex-specific differences exist in the expression of various KDMs within thyroid cancer tissues, providing insights into the potential role of epigenetic regulation in the sexual dimorphism observed in thyroid cancer.

  • A statement of the rationale of why these particular KDMs (KDM5C, KDM5D, and KDM6A) were selected for detailed analysis over others is needed.

While previous studies have suggested that sex differences in cancer may arise through the effect of circulating sex hormones, it has also been suggested that sex bias is derived from genetic and chromosomes independent of sex hormones. For example, the X linked lysine demethylase 6A and 5C, as well as Y-linked paralogs lysine demethylase 5D may be regulators of incidence and prognosis for sex-specific cancer [11, 12].

Tricarico R, Nicolas E, Hall MJ, Golemis EA. X- and Y-Linked Chromatin-

Modifying Genes as Regulators of Sex-Specific Cancer Incidence and Prognosis.

Clin. Cancer Res. 2020;26:5567–78. https://doi.org/10.1158/1078-0432.CCR-

20-1741.

Dunford A, Weinstock DM, Savova V, Schumacher SE, Cleary JP, Yoda A, et al.

Tumor-suppressor genes that escape from X-inactivation contribute to cancer

sex bias. Nat. Genet. 2017;49:10–6. https://doi.org/10.1038/ng.3726.

  • While the authors identified sex-specific expressions and localizations of KDMs, they may go and discuss the potential biological mechanisms being affected.

            We thank the reviewer for this comment. We have reworked the discussion and have incorporated additional insights into possible biological mechanism of KDMs (see the reference to role of KDM6A/UTX as an epigenetic contributor regulating chromatin accessibility and gene expression vital for NK cell homeostasis and effector function in a sex-specific fashion.

  • A more in-depth discussion needed, where more comparisons need to be made with existing literature. For example, highlighting how this study's findings align with or differ from previous research and what these differences might suggest about the underlying biology. Also, the discussion sometimes jumps between topics without clear transitions, making it challenging to follow the argument's thread.

We thank the reviewer for this comment. We have now included additional references to the literature relevant to our findings. We also have reworked the flow of the discussion with initial part referring to results relevant to gene expression (along with relevant references) and later part refers to protein localization.

  • Reference numbers should be placed in square brackets throughout the manuscript.

We have made the recommended change to the reference formatting.

Specific comments and questions

Please remove the extra period in the end of the title.

Unexpected space in line 53, 58, 425 and more…

We have made these changes in the manuscript and ensured all spelling mistakes and formatting issues are addressed.

Reviewer 3 Report

Comments and Suggestions for Authors

Dear authors,

The manuscript “Sex-Specific Expression of Histone Lysine Demethylases (KDMs) in Thyroid Cancer”, cancers-2923192, describes and compares expressions of sex-specific genes in women and men during the development of thyroid carcinogenesis. It was demonstrated that normal thyroid tissue has a sex-specific molecular signature, and the development of thyroid cancer is associated with differential expression of sex-biased genes. The paper is written in an easy-to-follow way, has some interesting noticing, and with a few minor corrections, I recommend it for publication.

Minor:

1.        How were the tissue samples collected and preserved (line 106)? Were the resected tissue samples immediately frozen in liquid nitrogen after surgery and stored at -80°C for further use, or did you extract RNA/DNA from FFPE samples? Please add the explanation in Section Quantitative real-time PCR (Q-RT-PCR) of the manuscript.

2.        What gene was used as an endogenous control? It is written that the relative fold gene expression of samples was calculated using the Delta-Delta Ct method (line 118).

3.        The information on the number of TC tissue microarray slides used should be added to the immunohistochemistry section of the manuscript.

4.        The presentation of Fig. 3 should be improved: the y-axes should be renamed (e.g., mRNA relative fold change), information about applied statistical tests and transformations should be placed in the figure legend, and terms such as columns B, C.. should be removed from the figure.

5.        Figure 4B, presented in this way, is confusing. The most important is that it misses explanation in the body text of the manuscript. Please describe how you got the figure, what the symbols “co/c (1 or 2)” represent, what the figure represents, and what the "wheels" are.

6.        As the authors explained (line 134–140), IHC staining should be presented as IHC scores with distinct values in a range of 0–16. But, in Tables 3 and 4, expressions of KDM proteins in thyroid tissue samples are presented via numbers±SD (e.g., 5±1.3). Can you explain the divergence between explanation and presentation of data? Please add the explanation of the numbers in the tables under the tables. In addition, in the text of the manuscript, IHC staining is presented via positive and negative. What is a positive result? Any value >0. If so, the explanation should be added to the text.

7.        How did you set the threshold for mRNA fold change (line 277)? Why did you select 3-fold? I suggest you apply a ROC analysis to elucidate the cut-off value to classify cases as tumors.

8.        The results of mRNA expression in selected tissue samples should be presented in a figure. Differences, (if any) would be much more easily noticed. The figure should present the results of mRNA expression in males and females, as well as differences (if any) in the KDMs mRNA fold changes in adenoma and carcinoma.

9.        Do you have information about the KDMs cell localization in adenoma? What is the KDMs cell localization in FTC and PDTC?

10.   The first sentence of the conclusion (line 442-443) should be rewritten as progression and recurrence was not analyzed in this manuscript.

11.   There are some typographical mistakes (e.g., lines 96, 158, 344, 442) that should be corrected.

Comments on the Quality of English Language

English language is fine, minor typographical mistakes required

Author Response

Dear authors,

The manuscript “Sex-Specific Expression of Histone Lysine Demethylases (KDMs) in Thyroid Cancer”, cancers-2923192, describes and compares expressions of sex-specific genes in women and men during the development of thyroid carcinogenesis. It was demonstrated that normal thyroid tissue has a sex-specific molecular signature, and the development of thyroid cancer is associated with differential expression of sex-biased genes. The paper is written in an easy-to-follow way, has some interesting noticing, and with a few minor corrections, I recommend it for publication.

Minor:

  1. How were the tissue samples collected and preserved (line 106)? Were the resected tissue samples immediately frozen in liquid nitrogen after surgery and stored at -80°C for further use, or did you extract RNA/DNA from FFPE samples? Please add the explanation in Section Quantitative real-time PCR (Q-RT-PCR) of the manuscript.

Response. Thyroid tissue samples were collected during surgery, immediately placed in RNA later solution and stored at -80°C for further use. 

  1. What gene was used as an endogenous control? It is written that the relative fold gene expression of samples was calculated using the Delta-Delta Ct method (line 118).

The 18S (NR_003286) was used as control for amplification of target gene by real-time-PCR. Each sample was run in triplicate.

  1. The information on the number of TC tissue microarray slides used should be added to the immunohistochemistry section of the manuscript.

KDM protein expression and localization by immunostaining were examined on 3 tissue microarray slides each containing 80 thyroid tissue samples from 60 female and 20 male patients. On each slide there were 10 samples from normal thyroid tissue, 20 from follicular cancer (FC), 44 from papillary thyroid cancer (PTC) and 6 from poorly differentiated thyroid cancers (PDTC).

  1. The presentation of Fig. 3 should be improved: the y-axes should be renamed (e.g., mRNA relative fold change), information about applied statistical tests and transformations should be placed in the figure legend, and terms such as columns B, C.. should be removed from the figure.

We modified Fig.3 as requested.

Modified Figure 3

Welch’s t-test confirmed over-expression of KDM5C in female TC [p=0.000 (t=26.3)]; overexpression of KDM5D in male TC [p=2.440e-208 (t=-58.48)]; and overexpression of KDM6A in female TC [p=0.000 (t=16.55)].

  1. Figure 4B, presented in this way, is confusing. The most important is that it misses explanation in the body text of the manuscript. Please describe how you got the figure, what the symbols “co/c (1 or 2)” represent, what the figure represents, and what the "wheels" are.

We agree with reviewer’s opinion. Figure 4B is confusing and does not provide critical information. We modified Figure 4 and removed Figure 4B.

Figure 4

  1. As the authors explained (line 134–140), IHC staining should be presented as IHC scores with distinct values in a range of 0–16. But, in Tables 3 and 4, expressions of KDM proteins in thyroid tissue samples are presented via numbers±SD (e.g., 5±1.3). Can you explain the divergence between explanation and presentation of data? Please add the explanation of the numbers in the tables under the tables. In addition, in the text of the manuscript, IHC staining is presented via positive and negative. What is a positive result? Any value >0. If so, the explanation should be added to the text.

We modified tables 3 and 4. The numbers in the table represent the average IHS scores and standard deviation in examined samples (normal thyroid, follicular cancer, papillary cancer and poorly differentiated thyroid cancers). Any value >0 was considered positive. For comparison of IHS scores between normal thyroid and thyroid cancers we performed Student T test. Results were considered significantly different with p=value less than 0.05.   

Table 3. Expression of KDM Proteins in Thyroid Tissue Samples from Female Patients

Histology

Immunostaining score

KDM5C

(average ± SD)

KDM5D

(average ± SD)

KDM6A

(average ± SD)

Normal

FTC

PTC

PDTC

2.25±1.2

1.92±2.0

3.0±3.2

2.0±2.7

0

0

0

0

9.75±1.5

5.1±2.9

10.7±3

6.25±2

Table 4. Expression of KDM Proteins in Thyroid Tissue Samples from Male Patients:

Histology

Immunostaining score

KDM5C

(average ± SD)

KDM5D

(average ± SD)

KDM6A

(average ± SD)

Normal

FTC

PTC

PDTC

1.5±1.3

1.5±1.5

1.7±1.4

1.5±0.7

10.0±2.4

4.7±4.1

2.3±2.7

2.0±0

7.0±1.5

6.2±4.5

7.5±2.8

7.0±1.4

  1. How did you set the threshold for mRNA fold change (line 277)? Why did you select 3-fold? I suggest you apply a ROC analysis to elucidate the cut-off value to classify cases as tumors.

We set a 3 fold threshold for mRNA fold change to maintain consistent approach with previously published study examining sex-biased gene expression in 13 different type of cancers (ref.13; Yuan Y, Liu L, Chen H, Wang Y, Xu Y, Mao H, Li J, Mills GB, Shu Y, Li L, Liang H 2016 Comprehensive Characterization of Molecular Differences in Cancer between Male and Female Patients. Cancer Cell 29:711-722.).

In that study, the propensity score algorithm was applied to identify the genes that show significant differences between male and female patients.

  1. The results of mRNA expression in selected tissue samples should be presented in a figure. Differences, (if any) would be much more easily noticed. The figure should present the results of mRNA expression in males and females, as well as differences (if any) in the KDMs mRNA fold changes in adenoma and carcinoma.

During preparation of this manuscript we have been trying to present results of mRNA analysis in form of figure. However this approach led to generation of 6 figures, showing in most cases no significant changes and taking more than 2 pages. So we switched to table format, demonstrating the most noticeable finding - a sex-specific expression of KDM5D.    

  1. Do you have information about the KDMs cell localization in adenoma? What is the KDMs cell localization in FTC and PDTC?

In adenomas the intracellular localization of KDMs was similar to those observed in normal thyroid tissue. We examined only 5 cases of FA (3 female and 2 male). Taking into consideration high morphological heterogeneity of thyroid adenomas (micro-follicular, macro-follicular, solid/trabecular, with papillary hyperplasia and other) and importance of differential diagnosis between follicular adenomas and follicular cancer we are planning further investigation of KDMs in a series of thyroid follicular neoplasms, and did not include these data in this paper.

What is the KDMs cell localization in FTC and PDTC?

 In male patient the immunostaining scores for KDM5D were significantly higher in normal thyroid as compared to FTCs (p=0.0247) and PDTCs (p=0.0.0004). Nuclear KDM5D expression was found in all normal thyroid tissue samples. In cancers, nuclear KDM5D was detected in 1/6 (16.6%) FTCs, 2/6 (33.3%) PTCs and 0/2 (0%) PDTCs.

In female patients the immunostaining scores for KDM6A were higher in normal thyroid tissue than in FTCs (p=0.0015) and in PDTCs (p=0.0048). Analysis of KDM6A intracellular localization showed exclusively nuclear expression in 4/4 (100%) normal thyroid, but only in 3/14 (21.4%) FTCs and 0/4 (0%) PDTCs.

  1. The first sentence of the conclusion (line 442-443) should be rewritten as progression and recurrence was not analyzed in this manuscript.

The sentence was changed as following:

Normal thyroid has a sex-specific molecular signature, and development of thyroid cancer is associated with differential expression of sex-biased genes.

  1. There are some typographical mistakes (e.g., lines 96, 158, 344, 442) that should be corrected.

This has been addressed.

Comments on the Quality of English Language

English language is fine, minor typographical mistakes required
